Analysing the effectiveness of Twitter as an equitable community communication tool for international conferences

Duncan Niall W. 1 2 niall.w.duncan@gmail.com
Shean Russell 3
1 Graduate Institute of Mind, Brain and Consciousness, Taipei Medical University , Taipei , Taiwan
2 Brain and Consciousness Research Centre, TMU-Shuang Ho Hospital , New Taipei City , Taiwan
3 Center for Data Science, Washington State Department of Health , Tumwater, Washington , United States of America
Antoun Jumana
Electronic publication date: 2023 May 8
Publication date: 2023
Volume: 11
Electronic Location ID: e15270
Received 2022 Dec 23; Accepted 2023 Mar 30
Copyright: © 2023 Duncan and Shean
Copyright year: 2023
Copyright holder: Duncan and Shean
License: This is an open access article distributed under the terms of the Creative Commons Attribution License, which permits unrestricted use, distribution, reproduction and adaptation in any medium and for any purpose provided that it is properly attributed. For attribution, the original author(s), title, publication source (PeerJ) and either DOI or URL of the article must be cited.
License URL: https://creativecommons.org/licenses/by/4.0/

Keywords: Scientific networks, Communication, Equality, Science communication, Online interaction

Funding: Taiwan Ministry of Science and Technology 108-2410-H-038-008-MY2 & 110-2628-H-038-001-MY4 Taiwan Ministry of Education DP2-111-21121-01-N-05-03 This work was supported by grants from the Taiwan Ministry of Science and Technology (108-2410-H-038-008-MY2 & 110-2628-H-038-001-MY4) and the Taiwan Ministry of Education (DP2-111-21121-01-N-05-03) to Niall W. Duncan. The funders had no role in study design, data collection and analysis, decision to publish, or preparation of the manuscript.

==============================
Scientific conferences increasingly include online aspects. Some are moving to be entirely virtual whilst others are adopting hybrid models in which there are both in-person and virtual elements. This development of opportunities for people to attend conferences virtually has the potential to both reduce their environmental impact and to make access to them more equitable. An issue with virtual conference participation that has been raised, however, is that there is a reduction in informal communication between attendees. This is an important deficit as informal contacts play a significant role in both knowledge transmission and professional network development. One forum where some informal communication around conferences does occur is Twitter, with participation there being encouraged by some conferences. It is not clear, however, how effective Twitter is as a community communication tool in terms of equitable participation amongst conference attendees. To investigate this, we looked at Twitter usage surrounding four international conferences between 2010 and 2021. It was found that engagement with conference hashtags increased steadily over time, peaking in 2019. Users represented 9% of conference attendees and were primarily located in Europe and North America, communicating primarily in English (97% of tweets). Hub nodes within the interaction network were also primarily located in these regions. East Asia had fewer users than would be expected based on neuroscience publication numbers from that region. What users there were in East Asia were engaged with less than were users in other regions. It was found that the overall interaction network showed a rich-club structure, where users with more connections tend to interact more with others with similar connection numbers. Finally, it was found that users in Europe and North America tend to communicate with other users in their own regions whereas users in the rest of the world direct their interactions out of their region. These results suggest that although conference-related Twitter use has been successful to some degree in opening up access, there are some notable limitations in its usage that may mirror aspects of inequality inherent to in-person conferences. How to build equitable informal communication networks around virtual conferences remains a challenging question that requires further discussion.

Introduction

Conferences play an important role in the scientific process. They provide a venue for information dissemination and form part of the social structures that underlie the production of scientific knowledge and facilitate academic researchers’ career development (de Leon & McQuillin, 2020; Oester et al., 2017). These outcomes are achieved through both the formal aspects of conferences, such as talks and symposia, and the informal contacts between participants that can occur (Hansen & Budtz Pedersen, 2018). Historically, such scientific conferences have predominantly been organised as in-person events held at a single location. Problems with this model have, however, become increasingly discussed.

One such issue is inequality of access to, and representation at, in-person conferences. People from lower-income countries are often excluded due to issues of cost or problems obtaining visas (James, 2002; Velin et al., 2021). Likewise, early-career researchers and those from smaller institutions can also find themselves unable to afford conference attendance. At the same time, groups such as women, LGBT+ people, and ethnic minorities can be under-represented as speakers at conferences and on organising committees (Sarabipour et al., 2021; Wheaton, 2021). Together, these factors can mean that opportunities to express views and to develop networks can be concentrated around people from particular geographical locations and with similar experiences.

Another issue for traditional conferences is their negative environmental impact, driven primarily by greenhouse gas emissions from air travel (Hischier & Hilty, 2002; Burtscher et al., 2020). The estimated average CO2 emissions for a person travelling to an international conference are about three tonnes (Klöwer et al., 2020). This is equivalent to what a person in a low-income country would produce in a decade and is unsustainable under current conditions if global heating is to remain under 1.5 °C (Rae et al., 2022; UN, 2021).

A potential solution for these issues that has been increasingly highlighted is adding more online aspects to conferences. Suggested changes range from making traditional physical conferences hybrid events with simultaneous online components to the creation of conferences that are entirely virtual (Raby & Madden, 2021; Sarabipour, 2020). Experiences from conferences that moved online due to COVID-19 restrictions have provided evidence for the efficacy of these changes, lending support to their wider acceptance (Niner & Wassermann, 2021; Sarabipour, 2020). In particular, access for previously under-represented groups was improved for virtual conferences compared to in-person events in previous years (Niner & Wassermann, 2021; Sarabipour, 2020). At the same time, these virtual events remove the need for air travel and so almost entirely eliminate conference-related carbon pollution (Rae et al., 2022).

A shift online brings its own issues though (Levitis et al., 2021; Valenti et al., 2021). In particular, participants of online conferences report that the experience struggles to replicate the informal interactions that occur at in-person conferences (Niner & Wassermann, 2021; Roos et al., 2020; Raby & Madden, 2021). These are of key importance as evidence suggests that the development of personal networks that they foster may play a greater role than formal conference presentations in both knowledge transmission and career development (Harrison, 2010; Oester et al., 2017; Storme et al., 2017). Similarly, a range of evidence points to the importance of direct communication for knowledge transmission and for the development of collaborative research.

One service that has been put forward as providing a medium for direct, informal online conference conversations is Twitter. Many conferences promote Twitter use with conference specific hashtags and, in some cases, integrate Twitter messages with conference presentations or discussions (Bombaci et al., 2016; Callister et al., 2019; McKendrick, Cumming & Lee, 2012). There is also an existing body of scientists who use Twitter to communicate about their work with other scientists and the wider community (Bex, Lundgren & Crippen, 2019; Leigh et al., 2021; López-Goñi & Sánchez-Angulo, 2018).

It is not clear, however, how effective Twitter is as a medium for informal communication around conferences. For it to be effective, it should allow engagement from the full range of people that participate in those events. This would mean that conversations about the scientific content of the conference would have the potential to reach any participant that wants to see them. As a further element of their effectiveness, the online informal networks that arise around conferences on Twitter should not exclude particular groups, nor should they reflect the power-imbalances in access and influence already seen in academic science (Sarabipour et al., 2021; Velin et al., 2021).

To investigate how effective Twitter is as a community communication tool in terms of equitable participation amongst conference attendees, we looked at its use across four international neuroscience conferences. These include two general neuroscience conferences and two neuroimaging specific ones. Firstly, engagement with each conference’s Twitter hashtag between 2010 and 2021 was analysed in terms of user numbers and user interactions, giving an overall view of participation by the conference communities. Following this, the number of users across different countries was investigated to establish their geographical distribution. User locations were then further analysed to describe the interaction networks that exist around the conferences and identify patterns of influence within them. Together, these analyses were used to establish who is contributing to the online conversation and whether access to this conversation appears to be geographically equitable. Other forms of equity, such as gender and within-region ethnicity differences, could not be investigated with the publicly available information used.

Methods

Ethical considerations

The analysis was conducted using information made publicly available by individuals through their Twitter profiles or journal publications. The content of individual tweets was not analysed, with only general properties, such as numbers of tweets or the locations users enter in their profile, being of interest. As the use of Twitter comes with legal jeopardy in some countries, specific location information for users in those places was removed. All usernames in the data shared with this work were converted to random strings to further protect individual privacy. The study was deemed exempt from review by the Taipei Medical University Institutional Review Board (N202203175).

Conferences

Four international conferences in the brain science domain were included: The Organisation for Human Brain Mapping Annual Meeting (OHBM); The International Society for Magnetic Resonance in Medicine Annual Meeting (ISMRM); The Society For Neuroscience Conference (SfN); and the Federation of European Neuroscience Societies Forum (FENS). OHBM (~2,500 attendees) and ISMRM (~5,000 attendees; (ISMRM, 2020)) were included as two specialist conferences directly familiar to the authors. SfN (~27,500 attendees; (Society for Neuroscience, 2022)) and FENS (~6,500 attendees; (FENS, 2022b)) were included as more general brain science conferences that involve a wider range of attendees. Tweets from conferences between 2010 and 2021 were studied. The FENS conference is held biennially in even-numbered years. SfN was cancelled in 2020 and so no data from that year were available. For each conference, the hashtag studied was its acronym plus the year (e.g., #OHBM2020). This format is commonly promoted by conference organisers and has the advantage of being distinct enough to be unlikely to return unrelated tweets. A manual check of tweets was conducted to remove any that were obviously unrelated to the conferences (e.g., discussion of the East Anglian Fens under the FENS conference hashtags).

Tweet information

Individual tweets containing the relevant hashtags were automatically searched for using the Twint package (https://github.com/twintproject/twint) running on Python 3.8. Unique usernames for each hashtag were identified and the number of tweets using the relevant hashtag sent by each calculated. The numbers of “likes”, “retweets”, and replies for each user per hashtag were established. These were then summed to give an overall count of interactions that each user had. The proportion of Twitter users for each conference relative to the number of registered attendees was calculated for 2018. This year was chosen as the most recent year where all four conferences were held without COVID-19 related disruption.

The language of each tweet was predicted using the CLD3 model (Google, 2022) implemented in the pycld3 package (https://github.com/bsolomon1124/pycld3). Only tweets with six or more words were used, excluding any URLs. Tweets where the classification of the language had a certainty of less than 90% were not included in the estimates. The proportion of tweets that were classified as being in English vs. other languages was calculated.

User locations

The geographical location connected to each user was extracted from their profile. This information was not available for all users as adding location information to a profile is optional and not all users include it. Note also that since the information is entered by the user it reflects the location which they wish to share and may not be the location they are actually at.

Having automatically extracted tweet locations from the user profiles, these were then manually modified according to the following criteria: (1) Fictional or impossible locations were deleted (e.g., “Narnia”); (2) overly general locations were deleted (e.g., “Earth”); (3) specific street addresses were removed (e.g., “No. 80 Street, Glasgow, United Kingdom” changed to “Glasgow, United Kingdom”); (4) locations in non-Latin script were converted; (5) flag emojis or nicknames for locations were converted; and (6) city names in countries where Twitter is blocked were removed (e.g., “Xi’an, China” changed to “China”). Where more than location was given, the first was used.

Latitude and longitude for each location was established using the GeoPy package (https://geopy.readthedocs.io/en/stable/), interfacing with the Google Maps API. The same API was used to confirm the country in which each user was located. Countries were also split into general geographical regions, such as South East Asia, North America, and Africa (see Table S1 for details; note that there were insufficient tweets from the region to support a more fine-grained division of the African continent).

Publication locations

To investigate whether patterns of tweets could be explained by differences in the number of people in each country doing research related to the conferences, location information from a set of relevant journal articles was collected. The journals looked at were eNeuro, European Journal of Neuroscience, Frontiers in Human Neuroscience, Frontiers in Neuroscience, Journal of Neuroscience, Human Brain Mapping, Magnetic Resonance in Medicine, and NeuroImage. These journals are either the journals of the societies that organise the conferences studied or are closely associated with the community attending them. A search on Pubmed (https://pubmed.ncbi.nlm.nih.gov/) was conducted for each journal, limited to publication dates between 2010 and 2020. Pubmed searches were conducted with the PyMed package (https://github.com/gijswobben/pymed).

The country of origin for each article was based upon the first affiliation of the senior author where the affiliation of more than one author was listed. Items in a journal that did not have an author attached (e.g., notes from the journal editor, corrigenda) were removed. Similarly, affiliations where the country was not clear were excluded from the analysis. A mean of 3.8% of articles were excluded in this way (range = 0.3–10.9%; Table S2), leaving 51,483 with location information. Countries were assigned to geographical regions in the same manner as Twitter users (Table S1). Eleven countries from which there were Twitter users did not have any publications (15.7%; Table S3).

Interaction networks

Networks of interactions between different users were established based upon to whom a tweet was indicated as being a reply to and upon any usernames included in the body of a tweet. The users identified in this way overlap with those directly using the conference hashtags but also includes additional users. Each user was taken as a node and the presence of a reply or mention of a username taken as an edge. Unweighted edges were used so that multiple tweets within a single conversation did not bias any individual user’s relative contribution to the network. Separate networks were built that included either all users or only those users for whom location information was available. Networks were created and analysed using the NetworkX package (Hagberg, Swart & Chult, 2008).

Degree centrality was calculated for each network node. These were then used to establish the rich-club coefficient at each node centrality (Zhou & Mondragon, 2004). The presence of a rich-club organisation was tested through comparison to 500 randomised networks (Colizza et al., 2006). Such an organisation would indicate that nodes with a given degree are more connected to similarly well connected nodes than they are to nodes that are less well connected. A VoteRank algorithm was then applied to identify the 5% most important nodes in the network in terms of influence on information flow across it (Zhang et al., 2016). Both rich-club and high-importance nodes were localised to regions. Finally, whether users were more likely to interact with others inside the same region as them or with users in other regions was quantified as the ratio of intra- to extra-regional edges.

Statistical analysis

The activity of users in terms of the number of tweets sent and number of interactions per tweet were first compared between users who had location information and those who did not to ensure those with information were not unrepresentative of users as a whole. The same metrics from the full dataset were then compared between conferences and across years. These comparisons were made through pseudo-rank Kruskal-Wallis tests (Brunner et al., 2018) followed by Dunn’s post-hoc tests with FDR correction for multiple comparisons (using PyNonpar and scikit_posthocs packages, respectively). The statistical threshold for Kruskal-Wallis tests was p = 0.05 and q = 0.05 for Dunn’s tests.

The influence of geographical location on Twitter behaviour and engagement was then tested. Firstly, users numbers in each country were correlated with the number of neuroscience publications from that country. This was done using Spearman’s correlation after log-transformation of the user and publication numbers. Countries that had no publications were excluded from this analysis. Next, countries that had unusually low or high users relative to their publication numbers were identified by calculating the Mahalanobis distance of each from the log(user) × log(publication) distribution and identifying distances greater than a threshold of p = 0.005 from a χ2 distribution with two degrees of freedom. Finally, the number of tweets sent by each user and the number of interactions they had per tweet were then compared between the eight different geographical regions (Kruskal-Wallis & Dunn’s tests).

In a final step, the user interaction networks were analysed. Median node degree was first compared between the full network and the network for which location information was available to ensure that the latter can be considered representative (Kruskal-Wallis test). Following this, the number of top 5% nodes per region was compared to the number expected given the total number of nodes per region (G-test).

Results

Conference engagement

A total of 42,857 tweets from 11,988 users were sent using the conference hashtags, producing 427,383 interactions. Of the total number of users, 8,638 were unique (i.e., not making contributions to more than one conference or year). A total of 6,638 unique users had location information available (76.8%). Numbers for each conference are shown in Table 1. Taking 2018 as an example year, Twitter users represented 9.0% of conference attendees (FENS = 10.5%; ISMRM = 1.2%; OHBM = 19.5%; SfN = 4.7%).

Table 1 Summary statistics for each conference.

	Total users	Unique users	Tweets	Interactions	Location (%)	
Conference						
FENS	2,183	1,881	11,586	109,192	77.9	
ISMRM	698	564	2,666	22,462	75.4	
OHBM	3,227	2,272	16,249	186,267	77.8	
SfN	5,880	4,705	12,356	109,462	79.4	

Overall, each user sent a median of one tweet per conference (IQR = 1.0–3.0, range = 1–444) and had a median of five interactions per tweet (IQR = 1.6–12.0, range = 0–2,825). Users for whom location information was available sent a mean of 0.26 more tweets (Kruskal-Wallis H(df=1) = 21.77, p < 0.001) but had the same number of interactions per tweet ( H(1) = 2.56, p = 0.11). The difference in the number of tweets sent was deemed to have no practical significance and so users with location details were treated as directly comparable to those without.

The number of tweets sent by each user differed between conferences, with users of all conferences but SfN sending a median of two tweets each and SfN users sending one ( H(3) = 1,017.48, p < 0.001; Fig. S1A). The number of interactions per tweet also differed between conferences ( H(3) = 412.32, p < 0.001), with OHBM users having the most (M = 7.5, IQR = 3.0–15.0, n = 3,227) and ISMRM users the least (M = 3.8, IQR = 1.0–10.0, n = 698; Fig. S1B).

Looking at how conference engagement changed over time, the overall number of tweets sent increased each year, from 428 in 2010 to a maximum of 10,179 in 2019 (Fig. 1A). A dip in this positive trend occurred in 2020 and 2021. The number of tweets sent per user also generally increased across the time studied ( H(11) = 347.37, p < 0.001; Fig. 1B), with some fluctuations, as did the number of interactions per tweet ( H(11) = 6,505.21, p < 0.001; Fig. 1C). A drop in both the number of tweets sent per user and in interactions per tweet was seen in 2021. The pattern of increasing engagement followed by a drop-off in 2020 and 2021 was similar for each conference individually, with the exception of FENS where numbers remained high in 2020 (see Fig. S2 for a breakdown of engagement over time for each conference).

Figure 1 (A) Number of tweets sent per year increases to a maximum in 2019 then falls to a low in 2021. (B) The number of tweets sent per user fluctuates by year, generally increasing to a peak in 2020 and then dropping off. (C) Users have an increasing number of interactions per tweet by year, with a slight reduction in 2021. An asterisk (*) indicates a change relative to the prior year (q < 0.05).

The language of an average of 76.3% (SD = 9.6%) tweets across conferences and years could be classified with at least 90% accuracy. Of these, a mean of 97.8% (SD = 1.9%) were classified as being in English.

User location

The geographical distribution of users is shown in Fig. 2A. Users came from 70 different countries and were predominantly located in North America and in Europe (Fig. 2B).

Figure 2 (A) Locations for unique users are indicated by a red dot. Note that users who give only a country as their location or who have had their city information removed will all appear at the same point within the relevant country. (B) Number of unique users per region. (C) Number of users per country plotted against number of publications on a log-log scale. The best-fit line is shown with a 95% confidence interval. Countries where the number of users is unexpected given the number of publications are highlighted in a dark circle. Countries are shaded according to their region. (D) Number of tweets sent per user. (E) Number of interactions/tweet. An asterisk (*) denotes that the relevant metric for the marked region is lower than the region with the corresponding asterisk colour (q < 0.05)

User numbers for a country were correlated with its number of neuroscience publications (Spearman’s ρ(57) = 0.86, 95% CI [0.76–0.92], p < 0.001; Figs. 2C and S3). Indonesia, Kenya, Nigeria, and Venezuela had more users than would be expected given the number of publications originating there, whilst China, South Korea, and Taiwan had fewer. It should also be noted that interaction network nodes were located in 11 countries that had no publications at all (Table S3).

The number of tweets sent by users differed between regions ( H(7) = 185.31, p < 0.001), with users in Africa sending the most per person (M = 3.0, IQR = 1.0–6.0, n = 24) and users in SE Asia the least (M = 1.0, IQR = 1.0–3.0, n = 101; Fig. 2D). The number of interactions per tweet that a user had also differed between regions ( H(7) = 85.08, p < 0.001), with users in Africa having the most (M = 7.2, IQR = 4.0–12.1, n = 24) and East Asia the least (M = 2.0, IQR = 0.2–6.8, n = 108; Fig. 2E).

Interaction networks

The full interaction network consisted of 8,578 nodes and 13,116 edges. Of these, 5,514 nodes had location information available (64.3%), connected by 7,750 edges (59.1%). The network of nodes with locations available had a lower maximum degree (M = 1, IQR = 1–2, range = 1–494) than did the network of nodes without (M = 1, IQR = 1–2, range = 1–528; H(1) = 8.59, p = 0.003) but the median degree and interquartile ranges matched and so the networks were deemed broadly comparable. A rich-club organisation was identified in both the full network and in the network of nodes for which there was location information (Fig. S4).

The geographical distribution of the network is shown in Fig. 3A. The 5% most important nodes were primarily located in Europe and North America (Fig. 3B; see also Fig. S5 for the top 20% of nodes). This distribution was broadly proportional to the numbers expected given the total number of nodes per region (G = 12.07, p = 0.09; Fig. S6). Nodes within Africa, Europe, and North America communicated more with other nodes within their own region (Fig. 3C). In contrast, nodes in the remaining regions were more likely to communicate with nodes in other regions.

Figure 3 (A) Interaction network nodes (orange) and edges (blue). (B) Top 5% most important nodes for information transfer across the network are shown. Note that users who give only a country as their location or who have had their city information removed will all appear at the same point within the relevant country. (C) Ratio of intra-regional to extra-regional edges. Positive values denote a majority of intra-regional communication.

Interactive results

The data supporting the results presented can be explored in a Shiny app (https://russellshean.shinyapps.io/twitter_shiny/). There, readers can view user location patterns and interaction networks for specific conferences or years.

Discussion

The results present a mixed picture of Twitter engagement around the conferences studied. On the one hand, the number of tweets sent increased steadily, up to the falloff that occurred in 2020, with each tweet tending to be engaged with more over time also. This suggests that people were increasingly using the medium to create informal discussion around the events. On the other hand, the majority of users sent between one and three tweets, which may suggest a relatively shallow level of engagement. At the same time, the peak proportion of people engaging in this informal online communication via Twitter was only around a tenth of the actual conference attendees, which is comparable to previously reported Twitter engagement statistics (Holmberg et al., 2014; Neill et al., 2014). It should be noted though that additional work would be required to judge if the level of engagement observed on Twitter was comparable or not to how people engage with in-person conference activities.

The conferences of 2020 and 2021 were held virtually due to pandemic restrictions (SfN was cancelled in 2020 and held virtually in 2021). A notable decrease in conference engagement of Twitter was observed for these years. Although the results reported do not provide insight into the reasons for this drop-off, a number of speculative explanations could be suggested. The first explanation would be that a general decrease in attendance for these virtual events compared to the years prior meant there were fewer people sending tweets. That FENS 2020, for example, had 4,780 participants (FENS, 2022a) compared to 7,324 in 2018 (FENS, 2022b) suggests that this is likely to have played a role. This 35% decrease in attendance (assuming similar differences were seen for the other conferences) would not, however, fully explain the over 50% reduction in tweet numbers between 2019 and 2021. A second potential explanation may be that the virtual events engender less engagement from participants than in-person ones, and so they are less inclined to send tweets about them. This would fit with reports over the same period of fatigue and disengagement brought on by working online (Bennett et al., 2021; Bonanomi et al., 2021; Singh Chawla, 2021). Thirdly, the reduction in tweets sent during virtual conferences may reflect a general difference in behaviour for people viewing an event at their computer compared to being physically present. “Real-life” engagement may prompt them to share their experience more than does engaging entirely through the computer. Finally, it may be that virtual events are successful in opening access for a more geographically diverse set of attendees and so are including more people outwith the regions where Twitter usage is concentrated, leading to a decrease in the proportion of attendees sending tweets. The true explanation is likely to include aspects of all these scenarios, amongst others, and will require additional data to understand fully.

Twitter engagement was seen across the globe, although the majority of users were located in Europe and North America. This pattern is not unexpected given the distribution of scientists involved in neuroscience research, as shown through a comparison with the number of relevant articles published by researchers in each country. It also reflects longstanding patterns of participation in international conferences (Sarabipour, 2020; Velin et al., 2021). Exceptions to this pattern were seen for some countries, with Indonesia, Kenya, Nigeria, and Venezuela having more users than would be expected based on publication numbers and China, South Korea, and Taiwan having fewer. In the case of the countries with more users than expected, each country only had one or two publications and so their results should be treated with caution. China, South Korea, and Taiwan had considerably more publications and so the results in their case are likely robust. This relative under-representation is not unexpected in the case of China, where access to Twitter is limited to a privileged minority. No such restrictions apply in Taiwan or South Korea but in their cases social media services other than Twitter are more commonly used, meaning that researchers may be interacting in other ways.

As well as being fewer in number than would be expected, users in East Asia (i.e., China, Japan, South Korea, and Taiwan) sent fewer tweets and had fewer interactions with other users than did people in other regions. A similar effect was seen for users in South East Asia, although to a lesser degree. It is not clear what might be driving these differences but language may play a role. Many languages in these regions do not use Latin script and so the fact that the majority of tweets are in English may present a barrier to engagement. This language barrier has been noted as a reason for under-representation of East Asian scientists as speakers at conferences and may be being replicated in the virtual domain (Perez Ortega, 2020; Takemura, 2020).

An analysis of the conference interaction network showed that it follows a rich-club structure (Zhou & Mondragon, 2004). This means that users tend to interact more with other users with the same number of connections or greater. This same structure has been reported in formal scientific collaboration networks (Opsahl et al., 2008), which are typified by a small number of influential individuals (Goh et al., 2003; Holme et al., 2002). Looking in more detail at the users with most connections in the conference network showed them to be prominent researchers within their field or to be the official accounts of scientific societies or related organisations. This does not preclude, of course, that other users gain greater visibility through the Twitter network than they otherwise would (Leigh et al., 2021). A rich-club-type structure was also observed in patterns of regional interaction directions. Users in Europe and North America were more likely to interact with other people within their own region. This was also true for users in Africa, although the small sample from this region means there must be some uncertainty about the robustness of this result. In contrast, users in the other regions were more likely to direct their interactions outwards. This suggests a flow from already less well represented regions towards already relatively privileged areas. Together, these interaction patterns may point to inequalities in access to academic networks. This could have effects on relative career development and on the possibility to investigate particular scientific questions (Li et al., 2022; Morgan et al., 2018; Shu, Sugimoto & Larivière, 2021).

An additional factor relevant to the effectiveness of social media services in the context of organising global scientific networks is their status as private, for-profit companies (Bak-Coleman et al., 2021). As private enterprises, what can and cannot be said on each service is dictated by the corporation and not the community (particularly where companies are subject to direct state censorship). This could lead to skewed scientific discussions and the exclusion of some community members, either of which eventuality would reduce the effectiveness of the medium. At the same time, their for-profit nature creates an ethical question around building networks where participation requires people to sign up to a particular company’s terms of use. In doing so they are generally required to give away access to their personal data, creating a tension between the privacy rights of individuals and the potential costs of not being able to participate in the network. At the same time, some social media companies have been highlighted as having negative effects on different societies (Bradshaw & Howard, 2019; Rapp, 2021), resulting in a similar tension for those who may not wish to support the companies but do want to participate in the scientific network. These ethical issues are unlikely to have any simple solutions, and the costs may be unavoidable, but they would appear to merit serious consideration.

Limitations

This investigation has a number of limitations. Firstly, user locations are based on what they report in their profile and so may not be accurate. Similarly, only one location per profile was used and so locations may reflect where someone was born, for example, rather than where they are currently located when both these locations were entered in their profile. More exact locations could be obtained through geolocation of tweets, but this information is (rightly) not publicly available. Secondly, it is possible that, although manual screening of tweets was conducted, some tweets that were unrelated to the conferences were included in the analysis. This is particularly true in the case of the network analysis where users may make mistakes when “tagging” people into a conversation. The relative number of unrelated tweets that may have been included is not, however, likely to be large enough to substantively change the pattern of results. Thirdly, our focus on tweets that use the conference hashtags means that some using the full conference name or other such formulations could have been missed. This could influence the overall pattern of results if the likelihood for people to use other formulations differed between conferences or years. The focus on conference hashtags also means that the results could be influenced by trends in hashtag usage. Any such changes in general posting behaviour may merit investigation in the future. Fourthly, the content of tweets was not analysed and so the nature of interactions is not known. This may be relevant to the interpretation of, for example, the rich-club structure of interaction networks. Finally, although the conferences studied include a large number of participants and cover a wide range of sub-disciplines, the results may not generalise to conferences in other fields.

Conclusions

These results suggest that services such as Twitter can play a role in informal communication around conferences but that there are potential limitations to their effectiveness in this role. In particular, when considering the creation of networks around scientific events there is a need to consider geographical differences in online habits and access, as well as issues of language (Levitis et al., 2021). This is also likely to be true of services other than Twitter and should be directly investigated in those different contexts.

Supplemental Information

Supplemental Information 1 Tweets and interactions per conference.

Number of (A) tweets sent and (B) interactions per tweet that each user had for each of the four conferences studied.

Click here for additional data file.

Supplemental Information 2 Tweets per year.

Number of tweets sent per year for each conference separately, normalised by the total number of tweets sent for that conference.

Click here for additional data file.

Supplemental Information 3 Publication and user numbers.

Publication and user numbers for each country plotted in original space. The main figure excludes the USA, with that country shown in the inset. Countries where the number of users is unexpected given the number of publications are highlighted in a dark circle. Countries are shaded according to their region.

Click here for additional data file.

Supplemental Information 4 Interaction network rich-club.

Rich-club coefficients (dark blue) at subsequent node degree centralities for (A) the whole network; and (B) the network composed of only nodes with location information. Rich-club coefficients from 500 randomised networks are shown in light blue, along with shaded 95\% confidence intervals.

Click here for additional data file.

Supplemental Information 5 Top 20% of nodes in the interaction network.

Click here for additional data file.

Supplemental Information 6 Important nodes per region.

Number of top 5% important nodes per region. Expected numbers were important nodes to be distributed evenly across regions are shown with dark lines.

Click here for additional data file.

Supplemental Information 7 Region assignment for each country.

Click here for additional data file.

Supplemental Information 8 Publications with location information from each journal.

Click here for additional data file.

Supplemental Information 9 Countries with Twitter users but no publications.

Click here for additional data file.

The authors would like to thank Dr Tzu-Yu Hsu for their input. The colour palettes used in making the figures were taken from Blake R Mills’ “MetBrewer” (https://github.com/BlakeRMills/MetBrewer).

Additional Information and Declarations

Competing Interests

Author Contributions

Human Ethics

Data Availability

The authors declare that they have no competing interests.

Niall W. Duncan conceived and designed the experiments, performed the experiments, analyzed the data, prepared figures and/or tables, authored or reviewed drafts of the article, and approved the final draft.

Russell Shean analyzed the data, authored or reviewed drafts of the article, interactive visualisation, and approved the final draft.

The following information was supplied relating to ethical approvals (i.e., approving body and any reference numbers):

The Taipei Medical University Institutional Review Board deemed the study exempt from review (N202203175).

The following information was supplied regarding data availability:

The data and code are available at OSF: Duncan, Niall W. 2023. “Twitter as a Community Communication Tool for International Conferences.” OSF. March 4. doi: 10.17605/OSF.IO/46WH2.

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
