# Peer review of "Analysing the effectiveness of Twitter as an equitable community communication tool for international conferences"

_PeerJ, doi:10.7717/peerj.15270_

## Round 0.1 · original submission · Minor Revisions

I would like to thank the authors for their meticulous description of the research methodologies. However, few improvements and clarifications are needed before a final decision is made. The reviewers have also added their concerns.

The main concerns are:

1- The research question is not clearly worded. The manuscript title is misleading and does not reflect the research question and results
2- The manuscript needs to be reviewed by an English editor.
3- The relevance of the publications and their locations to the research question is it clear to me. The selection of journals is not valid.
4- did you manually check all this huge number of tweets?

I have added an annotated manuscript for more detailed comments.

Reviewer 1 ·

Basic reporting

In this work, Duncan and Shean analyze over 42,800 tweets from four international conferences over the period of a decade to examine twitter engagement trends. This is a valuable study on role of social media, specifically Twitter in scientific interactions during and after conferences. I applaud the authors for conducting this study and for their clear and informative visuals and discussion.

Experimental design

The authors have explained their experimental design sufficiently well.

Validity of the findings

All underlying data have been provided and are robust, statistically sound, and controlled.

Additional comments

The authors note that they analyzed tweets that included the conference hashtag. Could there be a substantial number of tweets that instead only had the conference full name or abbreviated name only that are excluded from the present study? If so this could be mentioned in the Discussion section.

Did the authors analyze only the tweets with the respective conference hashtag or did they also included tweet replies to these tweets (replies without the conference hashtag) in their analysis?

Figure 3 shows Top 5% most important nodes for information transfer across the network are shown. Could the authors please provide a supplementary figure showing all important nodes or the top 20% of the important nodes?

·

Basic reporting

This paper provides a careful analysis of the twitter discussion generated by four prominent neuroscience conferences over the span of several years. This kind of work is important for us to understand the interaction of social media and conferences, and also the impact of virtual conferences. I particularly liked the analysis of the geographic patterns showing that interactions prioritized US/European areas. I also appreciate the careful wording of the paper which avoids inappropriate causal language.

Experimental design

The article is appropriate to the journal and the questions are meaningful and well designed. The methods are well described. I also commend the authors for making their analysis available through a shinyapp.

Validity of the findings

There are a few areas of this article that could be improved. Given that different conferences had different patterns of cancelling or going virtual, it seems important to see a breakdown of tweets broken down by conference, which perhaps could go into a supplemental. This would clarify to what degree the dropoff in tweets is related to going virtual or cancelling conferences.

I also think this paper is missing some obvious data that seems critical for supporting any links between virtual conferences and a reduction in twitter interaction. 2022 saw a return to in-person events, so an analysis of 2022 tweets would seem like an easy way to test this given that the tools are already in place.

I am also wondering if the dropoff in tweets is related to different patterns of hashtag usage over time rather than a reduction in tweets. Could a baseline hashtag usage analysis be used to understand if there is a general trend that could explain the pattern in Figure 1?

Additional comments

Nodes within Africa communicated more with themselves than outside, but this isn’t mentioned in the abstract alongside Europe and North America.

It should be mentioned that the Charts and Stuff pane of the Shinyapp was not working when I tried it.

It is stated in the discussion that the # of tweets increased over time, but this is not accurate since they peaked in 2019 and then declined sharply.

Several possible explanations are given for the decline in tweets with virtual conference attendance, and these are all plausible but I wonder if another one is that people sitting at their computers may actually tweet less than people in person because their mode of interaction with the computer is different. This seems paradoxical but when I am in person at a conference I feel more inclined to share the conference, while when I’m at a computer that impulse isn’t there. The authors may disagree and not choose to include this of course, as it is just speculation.

---

## Round 0.2 · Minor Revisions

Dear Authors

Thank you for the effort to revise and address the comments.

However, we are still unclear on several points

First, what is the precise question being asked? what is the research question in one statement that starts with the research aim to .....I still need further clarification to gain a full understanding of the research questions and objectives, thus deciding on the title. I can not see the research questions clearly and robustly in one statement. You have described the methods but not the research question in the last paragraph of the introduction. How would you say the communication tool was effective? How would you judge its effectiveness? How did you operationalize the effectiveness of the communication tool in the research analysis? Effectiveness could be measured in different ways. It could be the ability of the Twitter community to lead to more publications, or more attendance at the next conference, or knowledge dissemination. From my understanding, you were trying to measure whether Twitter was able to create a geographically diverse community. Then, this should be clear in the title.

Second, country user numbers and neuroscience publications seemed to correlate with the first statement supporting this trend. However, the next sentences indicated that there were outliers! What statistical analysis did you use to come to the result of the last statement about the actual users being more or less than expected? How did you measure "the expected"? Why did you check some countries when the overall statement is that there is a correlation?

---

## Round 0.3 · Minor Revisions

Dear authors

Thank you for your response. I would appreciate your patience and understanding to make the manuscript more robust.

I still have two minor comments:
1- Would you agree to change the following:
a- the title to "Analyzing the effectiveness of Twitter as a community communication tool for equitable participation in international conferences."
b- Abstract: Line 22-23. Change the statemen to "This study aims to study the effectiveness of Twitter as a community communication tool for equitable participation among conference attendees." or any other variation that keeps the same meaning.
c- lines 90, change "to investigate this question of effectiveness of Twitter on equitable participation" or any other variation.
c- Lines 100-101, the sentence structure and thus meaning is not clear. I suggest omitting the statement.

---

## Round 0.4 · accepted · Accept

I would like to thank the authors for their patience and cooperation throughout the review process. The manuscript is currently in a robust state, with a level of clarity that leaves no room for ambiguity.